# NK cells inhibit *Plasmodium falciparum* growth in red blood cells via antibody-dependent cellular cytotoxicity

Gunjan Arora[1], Geoffrey T Hart[1,2], Javier Manzella-Lapeira[1], Justin YA Doritchamou[3], David L Narum[3], L Michael Thomas[1†], Joseph Brzostowski[1], Sumati Rajagopalan[1], Ogobara K Doumbo[4], Boubacar Traore[4], Louis H Miller[5], Susan K Pierce[1], Patrick E Duffy[3], Peter D Crompton[1], Sanjay A Desai[5], Eric O Long[1]*

[1]Laboratory of Immunogenetics, National Institute of Allergy and Infectious Diseases, National Institutes of Health, Rockville, United States; [2]Department of Medicine, University of Minnesota, Minneapolis, United States; [3]Laboratory of Malaria Immunology and Vaccinology, National Institute of Allergy and Infectious Diseases, National Institutes of Health, Rockville, United States; [4]Malaria Research and Training Centre, Department of Epidemiology of Parasitic Diseases, International Center of Excellence in Research, University of Sciences, Techniques and Technologies of Bamako, Bamako, Mali; [5]Laboratory of Malaria and Vector Research, National Institute of Allergy and Infectious Diseases, National Institutes of Health, Rockville, United States

*For correspondence: elong@nih.gov

Present address: †BioMedicine Design, Pfizer Inc., Cambridge, United States

**Abstract** Antibodies acquired naturally through repeated exposure to *Plasmodium falciparum* are essential in the control of blood-stage malaria. Antibody-dependent functions may include neutralization of parasite–host interactions, complement activation, and activation of Fc receptor functions. A role of antibody-dependent cellular cytotoxicity (ADCC) by natural killer (NK) cells in protection from malaria has not been established. Here we show that IgG isolated from adults living in a malaria-endemic region activated ADCC by primary human NK cells, which lysed infected red blood cells (RBCs) and inhibited parasite growth in an in vitro assay for ADCC-dependent growth inhibition. RBC lysis by NK cells was highly selective for infected RBCs in a mixed culture with uninfected RBCs. Human antibodies to *P. falciparum* antigens PfEMP1 and RIFIN were sufficient to promote NK-dependent growth inhibition. As these results implicate acquired immunity through NK-mediated ADCC, antibody-based vaccines that target bloodstream parasites should consider this new mechanism of action.
DOI: https://doi.org/10.7554/eLife.36806.001

## Introduction

*Plasmodium falciparum (P. falciparum.)*, the causative agent of malaria, expresses proteins that are displayed at the surface of infected red blood cells (RBCs). Some of these proteins promote sequestration of *P.f.*-infected RBCs (iRBCs) through adhesion to vascular endothelial cells (*Miller et al., 2002*). Humans living in malaria-endemic areas generate, over years of repeated infections, antibodies (Abs) to *P.f.* proteins that contribute to the gradual protection from malaria symptoms (*Boyle et al., 2015*; *Bull and Marsh, 2002*; *Cohen et al., 1961*; *Mayor et al., 2015*; *Tran et al., 2013*). One of the main objectives in malaria research is to define the mechanisms by which naturally acquired Abs provide protection (*Cohen et al., 1961*; *Crompton et al., 2014*). Acquired immunity

**eLife digest** Malaria is a deadly disease caused by a parasite transmitted by mosquitoes. The parasite infects red blood cells, causing fever with flu-like symptoms. In some people, particularly pregnant women and children, the disease may be very serious and even lead to death. An effective malaria vaccine is urgently needed because malaria parasites are developing resistance to current drugs.

People living in areas where malaria is common develop specific proteins called antibodies that protect them from malaria. Learning more about how the antibodies achieve this, could help to develop better vaccines. Scientists already know some antibodies bind to the malaria parasites and prevent them from entering red blood cells. Some vaccines have been based on these antibodies. Other antibodies bind to infected cells flagging them for destruction by cells of the immune system. Immune cells called natural killer cells can eliminate viruses or cancer cells this way, but it was not clear if they could also eliminate malaria parasite-infected red blood cells.

Now, Arora et al. show that natural killer cells can selectively destroy malaria-infected red blood cells flagged with antibodies from people who live in areas where malaria is common. In laboratory experiments, natural killer cells from US volunteers, who were never exposed to malaria, did not kill normal or malaria-infected red blood cells. Adding antibodies collected from malaria-resistant volunteers from Africa allowed these natural killer cells from unexposed people to selectively seek out and destroy malaria-infected cells and leave uninfected red blood cells intact.

Arora et al. also found that the antibodies from the malaria-resistant volunteers bound to parasite proteins on the surface of infected blood cells. The experiments suggest that vaccines designed to stimulate the production of antibodies to malaria proteins that are displayed on infected red blood cells, could destroy the parasite in infected people and help prevent disease and save lives.

DOI: https://doi.org/10.7554/eLife.36806.002

to malaria is complex as it requires a balance of parasite growth inhibition and control of inflammation (*Zhou et al., 2015*). Neutralizing Abs that prevent *P.f.* merozoite invasion of RBCs have been described (*Douglas et al., 2011*). However, as merozoites released from late-stage iRBCs rapidly invade uninfected RBCs (uRBCs), high antibody titers are likely needed for inhibition. Abs bound to iRBCs promote phagocytosis by myeloid cells, and Abs bound to merozoites activate the complement pathway (*Bouharoun-Tayoun et al., 1995*; *Boyle et al., 2015*; *Rowe et al., 1997*).

Natural killer (NK) cells constitute about 10% of peripheral blood lymphocytes. They kill virus-infected cells and tumor cells through engagement of an array of germ-line encoded co-activation receptors (*Bryceson et al., 2006a*; *Cerwenka and Lanier, 2001*). In addition to their innate ability to eliminate transformed and infected cells, NK cells perform Ab-dependent cellular cytotoxicity (ADCC) through the low-affinity IgG receptor FcγRIIIa (also known as CD16), thereby killing IgG-coated target cells and secreting pro-inflammatory cytokines such as IFN-γ and TNF-α. A clear role of NK cells in contributing to protection from malaria, and whether iRBCs could be eliminated through ADCC by NK cells, has not been established (*Wolf et al., 2017*). Earlier studies have described direct lysis of iRBCs by NK cells in the absence of Abs or Ab-dependent inhibition of *P.f.* growth by NK cells (*Mavoungou et al., 2003*; *Orago and Facer, 1991*). However, other studies have not confirmed such results (*Wolf et al., 2017*). Here, we present a detailed study of the activity of primary, unstimulated human NK cells mixed with RBCs, infected or not by *P.f.*, and evaluate the NK cell responses using several different quantitative assays. We found that IgG in plasma from subjects living in a malaria-endemic region in Mali bound to iRBCs and induced their rapid lysis through NK-mediated ADCC. Naturally acquired IgG specific for the major *P.f.* antigen PfEMP1 was sufficient to promote NK-dependent inhibition of *P.f.* growth in RBCs. Our results demonstrated that primary human NK cells alone are capable of controlling parasite growth in vitro in response to IgG from subjects exposed to malaria. This may represent an important component of Ab-dependent clinical immunity to *P.f.* blood-stage infection that could be exploited in the development of malaria vaccines.

## Results

### Primary human NK cells are activated by *P.f.*-infected RBCs in presence of plasma from malaria-exposed individuals

RBCs infected with *P.f.* strain 3D7 were enriched for the presence of knobs at the RBC surface (*Figure 1—figure supplement 1A*). Knobs are protrusions at the surface of iRBCs that appear at the trophozoite stage. iRBC cultures were enriched for the trophozoite stage by percoll-sorbitol gradient. Enrichment was confirmed by Giemsa stain (*Figure 1—figure supplement 1B*). A pool of plasma from malaria-exposed adults living in a high-transmission region of Mali (Mali plasma) was tested for the presence of Abs to the surface of *P.f.* 3D7-iRBCs at the trophozoite stage by flow cytometry. Adults at the Mali study site are considered 'semi-immune' to malaria, as they generally control parasitemia and rarely experience malaria symptoms (*Tran et al., 2013*). Abs in Mali plasma stained iRBCs but not uRBCs (*Figure 1A*). In contrast, Abs in a pool of serum from malaria-naïve US adults (US serum) did not bind to iRBCs any more than they did to uRBCs (*Figure 1A*). Binding of Abs in Mali plasma to iRBCs was confirmed by immunofluorescence microscopy (*Figure 1B*). Lower magnification images of mixed uRBCs and iRBCs showed that staining by Mali plasma was selective for iRBCs (*Figure 1—figure supplement 1C*).

We tested the reactivity of primary NK cells, freshly isolated from the blood of healthy malaria-naïve US donors, to iRBCs in the absence of Abs. NK cells did not degranulate during co-incubation with iRBCs, as monitored by staining with anti-LAMP-1 (CD107a) Ab (*Figure 1C and D*). As binding of FcγRIIIA to IgG alone is sufficient to induce activation of resting NK cells (*Bryceson et al., 2005*), IgG bound to RBCs has the potential to induce NK cell degranulation and cytokine production. We first tested stimulation of NK cells in the presence of a polyclonal serum of rabbits that had been immunized with human RBCs. Degranulation by NK cells occurred during incubation with iRBCs in the presence of rabbit anti-RBC Abs (*Figure 1C and D*). Notably, potent NK cell degranulation occurred with iRBCs in the presence of Mali plasma, whereas US serum induced degranulation in a very small fraction of NK cells (*Figure 1C and D*). NK cell expression of intracellular interferon (IFN)-γ and tumor necrosis factor (TNF)-α was also stimulated equally well by rabbit anti-RBC serum and Mali plasma, whereas US serum did not induce cytokine production (*Figure 1E and F* and *Figure 1—figure supplement 1D*). These results suggested that Abs from malaria-exposed individuals activate NK cells when bound to iRBCs, which results in NK cell degranulation and production of cytokines.

### Selective lysis of *P.f.*-infected RBCs by primary NK cells in the presence of plasma from malaria-exposed individuals

We next investigated whether NK cells could selectively lyse Ab-coated iRBCs without causing bystander lysis of uRBCs. uRBCs and iRBCs were labeled with either eFluor450 or eFluor670 dyes, which bind cellular proteins containing primary amines, and NK cells were labeled with the lipophilic dye PKH67. The three cell types were incubated together at equal numbers, and examined by live microscopy. Images were acquired in a temperature-controlled chamber every 30 s for several hours (*Videos 1* and *2*). Representative images taken at 0, 2, and 4 hr are shown in *Figure 2A*. Quantitative analysis of cell numbers, which were determined every minute, showed that all three cell types remained at a constant ratio when incubated with US serum (*Figure 2B*, left panel). In contrast, NK cells selectively lysed iRBCs in the presence of Mali plasma, leaving uRBCs intact (*Figure 2B*, right panel). A compilation of four experiments, each performed with NK cells from a different donor, showed iRBC lysis induced by Mali plasma but not US serum (*Figure 2C*, left panel), and selective lysis of iRBCs in the presence of Mali plasma (*Figure 2C*, right panel). The relative change in the frequency of uRBCs and iRBCs over 3 hr in the presence of US serum or Mali plasma is shown in *Videos 1* and *2*. We concluded that lysis of *P.f.*-iRBCs by NK cells in the presence of plasma from malaria-exposed individuals was efficient and specific, causing minimal bystander lysis of uRBCs.

### NK cells inhibit *P.f.* growth in RBCs in the presence of IgG from malaria-exposed individuals

The fraction of RBCs invaded by merozoites (also known as parasitemia) in *P.f.*-infected individuals typically ranges from 0.005 to 5% (*Gonçalves et al., 2014*). We tested the ability of NK cells to inhibit parasite growth in an RBC culture that was set at 1% parasitemia. As the ratio of NK cells to

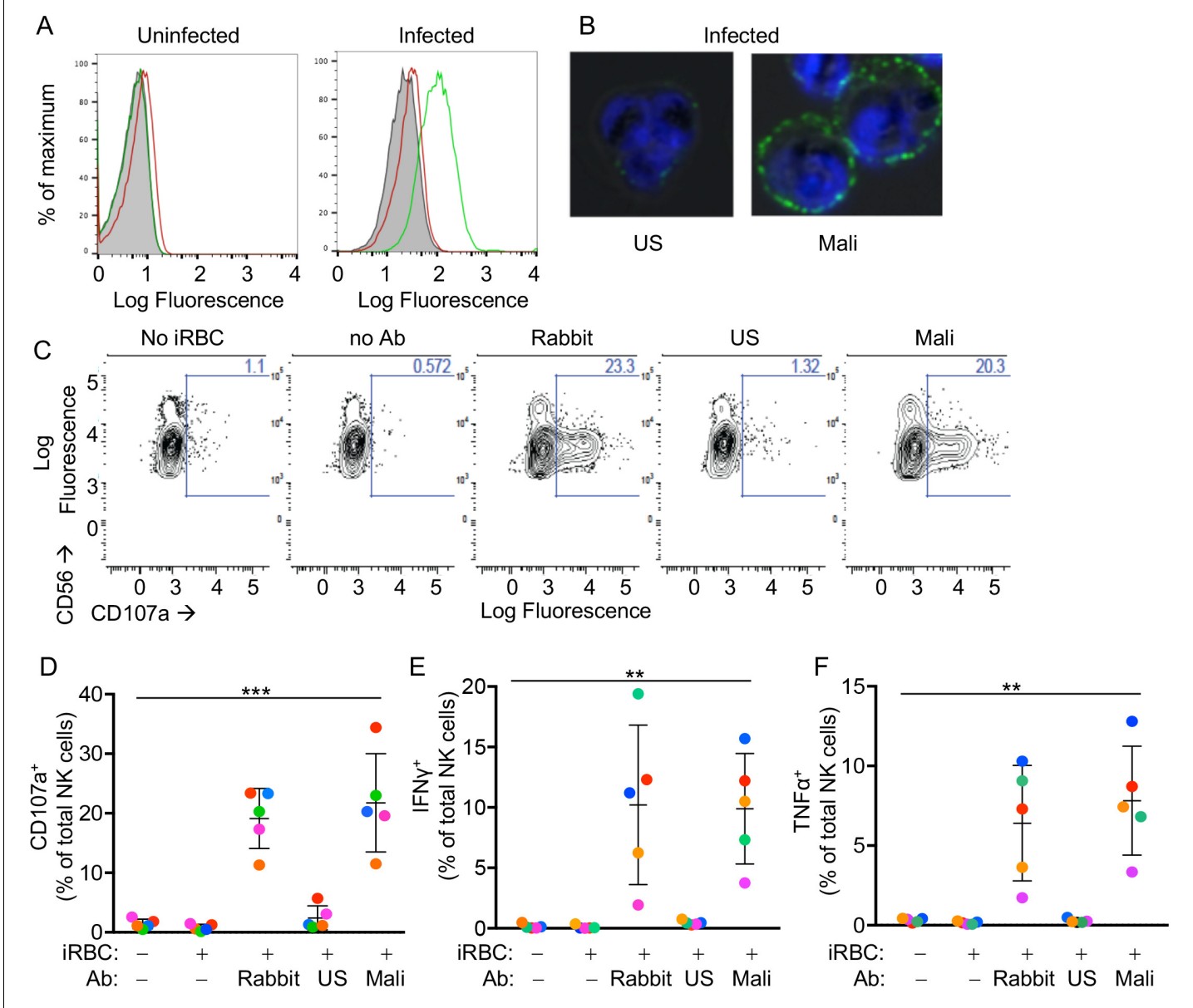

**Figure 1.** Primary human NK cells are activated by antibody-coated *P.f.*-iRBCs. (**A**) Uninfected RBCs (left) and trophozoite-stage iRBCs (right) were stained with serum pooled from US individuals (red) and plasma pooled from individuals living in a malaria-endemic region of Mali (green). Bound Abs were detected with AF488-conjugated anti-human IgG (H + L) antiserum. (**B**) Immunofluorescence images of iRBCs stained with DAPI (blue) and with either US serum or Mali plasma, as indicated. Anti-human IgG (H + L) antiserum labeled with AF488 (green) was used to detect Ab-coated RBCs. (**C**) NK cells stained with PE-Cy5.5-conjugated CD56 and PE-conjugated CD107a Abs. The fraction of CD107a$^+$ NK cells is indicated in each panel. (**D**) NK cell degranulation measured by CD107a Ab staining. NK cells alone or co-incubated with iRBCs, at a NK:RBC ratio of 1:1 for 4 hr, in the absence or presence of Abs, as indicated. Rabbit polyclonal anti-RBC serum (1:4000), US serum (1:10) and Mali plasma (1:10) were used. Circles indicate individual NK cell donors, each with its own color. Data from independent experiments are shown as mean ± SD (ANOVA, p=0.0009). (**E, F**) Intracellular staining of IFN-γ and TNF-α. Incubation conditions and Abs as in (**D**) (ANOVA, p=0.0061 for E, 0.0050 for F). Data from independent experiments are shown as mean ± SD.

DOI: https://doi.org/10.7554/eLife.36806.003

The following figure supplement is available for figure 1:

**Figure supplement 1.** Antibody-dependent cytokine response of NK cells in response to iRBCs.
DOI: https://doi.org/10.7554/eLife.36806.004

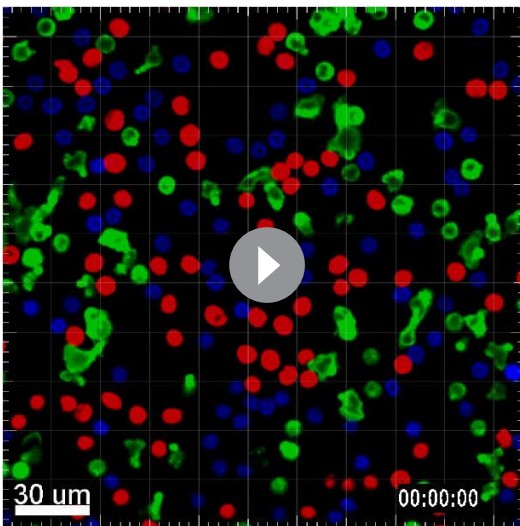

**Video 1.** shows a co-culture of NK cells, uRBC and iRBC in the presence of US serum, imaged for104 min.
DOI: https://doi.org/10.7554/eLife.36806.007

iRBCs was set at 3:1 and 1:1, NK cells were outnumbered by a 30 to 100 fold excess of uRBCs during incubation. Cultures were maintained for 48 hr before analysis (*Figure 2—figure supplement 1A*). Given that iRBC cultures were synchronized at the ring stage and enriched at the trophozoite-stage, a single round of RBC rupture and reinvasion of fresh RBCs by released merozoites occurred in the next ~18 hr. *P.f.* growth was determined by counting iRBCs in blood smears. In the absence of Abs, growth inhibition was 5.69 ± 11.53% at an E:T ratio of 3 (*Figure 2D*). A similar result was obtained in the presence of US serum (4.33 ± 12.15%; *Figure 2D*). In contrast, in the presence of Mali plasma, inhibition of parasite growth was 62.56 ± 15.41% at an E:T ratio of 3 (*Figure 2D*). Strong growth inhibition occured also at an E:T ratio of 1 (*Figure 2D*). A much reduced inhibition occurred with Mali plasma in the absence of NK cells (11.55 ± 1.99%), which could be due to Ab-mediated inhibition of merozoite reinvasion. We concluded that NK cells, in the presence of plasma from malaria-exposed individuals, are capable of inhibiting blood-stage *P.f.* growth even in the presence of a 100-fold excess of uRBCs. The results further suggested that maturation of trophozoites and schizonts into infectious merozoites was inhibited by NK-mediated ADCC toward iRBCs.

The standard growth inhibition assay (GIA) (*Malkin et al., 2005*) was modified to remove Abs that inhibit *P.f.* growth through neutralization of merozoites. NK cells were first co-incubated with trophozoite-enriched iRBCs for 6 hr, in the presence or absence of Mali plasma. Cultures were then washed to remove unbound Abs and soluble factors prior to addition of a 100-fold excess of fresh uRBCs. Cultures were further incubated for 16 hr to allow for a single round of merozoite release and reinvasion of uRBCs (*Figure 2—figure supplement 1B*). We refer to this assay for inhibition by NK-mediated ADCC as GIA-ADCC. Parasitemia at the end of the experiment was determined by flow-cytometry (*Figure 2—figure supplement 1C*). Inhibition of *P.f.* growth occurred in the presence of Mali plasma but not in the presence of US serum (*Figure 2E*). These results showed that inhibition of *P.f.* growth was due to Abs bound to iRBCs prior to the release of merozoites and the addition of uRBCs, confirming the role of NK cell-mediated ADCC.

As NK cell-mediated ADCC triggered by FcγRIIIa is dependent on binding to IgG, we tested whether IgG in Mali plasma was sufficient to promote NK-dependent inhibition of *P.f.* growth. IgG purified from US serum did not bind to uRBCs (*Figure 2—figure supplement 1D*) or to trophozoite-stage iRBCs (*Figure 2F*), whereas IgG purified from Mali plasma bound to iRBCs (*Figure 2F*) but not uRBCs (*Figure 2—figure supplement 1D*). In the GIA-ADCC, designed to exclude merozoite neutralization as the basis for inhibition, purified IgG from Mali plasma inhibited *P.f.* growth (37.59 ± 12.15% inhibition at an IgG concentration of 1.8 mg/ml) (*Figure 2G*). No inhibition was observed with IgG purified from

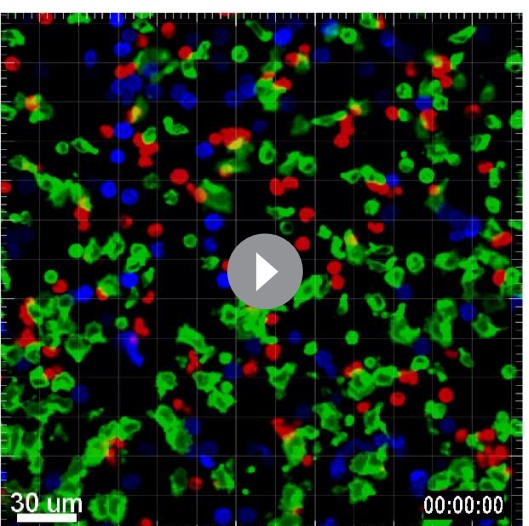

**Video 2.** shows a co-culture of NK cells, uRBC and iRBC in the presence of Mali plasma, imaged for 104 min.
DOI: https://doi.org/10.7554/eLife.36806.008

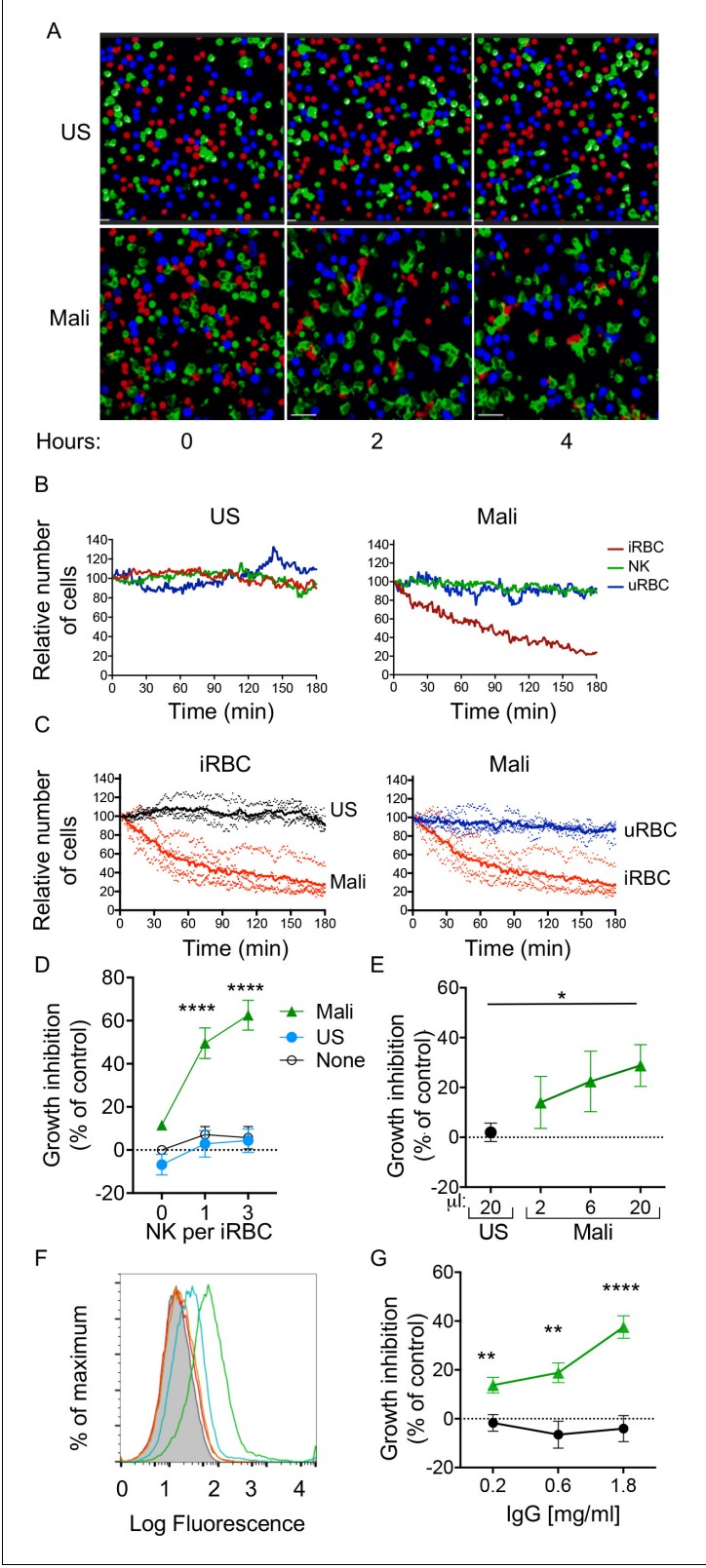

**Figure 2.** Selective lysis of *P.f.* 3D7-iRBCs and parasite growth inhibition by primary NK cells in the presence of immune plasma and IgG. (**A**) Live imaging of primary NK cells (green) co-incubated with uRBCs (blue) and iRBCs (red) at an equal ratio (1:1:1) in the presence of US serum (1:10) and of Mali plasma (1:10). Representative snapshots taken at time 0, 2, and 4 hr are shown. (**B**) Quantitative analysis of cell numbers in the cultures shown in

*Figure 2 continued on next page*

*Figure 2 continued*

(**A**) in a 3 hr period. Cell numbers were normalized to 100 at the start of image acquisition. (**C**) Composite display of 4 independent experiments, each carried out with a different NK cell donor (dotted lines). The mean is shown as a solid line (t test, p<0.0001). (**D**) Inhibition of parasite growth measured by counting blood smears of iRBCs. A parasite culture containing 1% iRBCs was incubated for 48 hr in the absence (open circles) or presence of US serum (closed circles) or Mali plasma (triangles). Growth inhibition is represented as percent decrease in parasitemia relative to a culture with no NK cells and no Ab. Error bars represent standard deviation of the mean from four independent experiments (ANOVA, p<0.0001 for no NK or US serum group compared with Mali plasma groups in presence of NK cells). (**E**) Parasite growth inhibition measured by flow cytometry. Enriched trophozoite-stage iRBCs were incubated with NK cells at an NK:iRBC ratio of 3:1 for 6 hr with either 20 µl US serum or increasing amounts of Mali plasma in a final volume of 200 µl. Cells were washed and incubated for another 16 hr with a 100-fold excess of uRBCs (relative to the iRBC input). Inhibition is expressed as a percent decrease in parasitemia relative to parasitemia in iRBC cultures incubated with NK cells in the absence of Abs (ANOVA, p=0.0294). (**F**) Staining of iRBCs with IgG affinity-purified from US serum at 0.2 (orange) and 0.6 mg/ml (red), or from Mali plasma at 0.2 (blue) and 0.6 mg/ml (green). (**G**) Growth inhibition assay performed as in (**E**) in the presence of purified IgG from US (black circles) or Mali individuals (green triangles) at the indicated concentrations (t test p(0.2) = 0.008; p(0.6) = 0.003; p(1.8) = 0.00007).
DOI: https://doi.org/10.7554/eLife.36806.005
The following figure supplement is available for figure 2:

**Figure supplement 1.** Assays for NK-dependent parasite growth inhibition.
DOI: https://doi.org/10.7554/eLife.36806.006

---

US serum (*Figure 2G*). The use of purified IgG eliminated the possibility that the difference observed between Mali and US individuals was due to properties unique to human plasma that were absent in human serum. These results demonstrated that IgG from malaria-exposed individuals promotes inhibition of *P.f.* growth in RBCs in the presence of NK cells.

## PfEMP1 is a major target of naturally acquired antibodies that induce NK-dependent lysis of iRBCs

The *P.f.* erythrocyte membrane protein 1 (PfEMP1), which mediates parasite sequestration through binding to vascular endothelial cells, is a major target of host Ab responses (*Bull and Marsh, 2002*; *Chan et al., 2012*). We used the parasite line DC-J, which lacks PfEMP1 expression (*Dzikowski et al., 2006*), to test the importance of PfEMP1 in promoting Ab-dependent NK cell activation. Staining of *P.f.* DC-J-iRBCs with Mali plasma gave a positive signal that was approximately one log less than staining of 3D7-iRBCs (*Figure 3A*), but greater than staining of *P.f.* DC-J-iRBCs with US serum (*Figure 3B*). Time-lapse imaging was used to monitor lysis of DC-J-iRBCs by NK cells in the presence of Mali plasma during co-incubation with an equal number of uRBCs (*Figure 3C*). NK cells did not lyse DC-J-iRBCs in the presence of Mali plasma (*Video 3*). Data from four experiments with NK cells from different donors indicated no significant decrease in iRBCs in the presence of Mali plasma compared to US serum over the course of 3 hr (*Figure 3D*). Therefore, residual Ab-binding in the absence of PfEMP1 (*Figure 3A*) was not sufficient, under the conditions used, to promote NK-mediated ADCC in the presence of Mali plasma.

We wanted to test whether the lack of lysis of *P.f.* DC-J-iRBCs by NK cells could perhaps be due to an intrinsic resistance of DC-J to NK-mediated lysis. To test it we used the rabbit anti-serum raised against human RBCs, which activated degranulation by NK cells in the presence of 3D7-iRBCs (*Figure 1C and D*). We further developed a quantitative RBC lysis assay based on hemoglobin (Hb) release into the supernatant. Maximum Hb release from RBCs was defined as Hb in detergent lysates of RBCs (*Figure 3—figure supplement 1A*). This control also served to compensate for the loss of Hb during *P.f.* development in RBCs, as the parasite digests some of the Hb to produce hemozoin. Severe damage to RBCs, as determined by Hb release, occurred at NK cell to RBC ratios of 3:1 and 10:1, after a 5 hr incubation with 3D7-iRBCs in the presence of rabbit anti-RBC Abs (*Figure 3—figure supplement 1B* and *Figure 3—figure supplement 1C*). At an NK cell to iRBC ratio of 5:1, 47.16 ± 8.76% of total Hb content was released (*Figure 3E*). The timing of eukaryotic evolution Does a relaxed molecular clock reconcile proteins and fossils? A small amount of Hb was released in the absence of NK cells (1.36 ± 0.51%) and in the absence of rabbit anti-RBC Abs (4.38 ± 1.65%) (*Figure 3E*). Similar data were obtained with uRBCs under the same conditions, where 47.15 ±

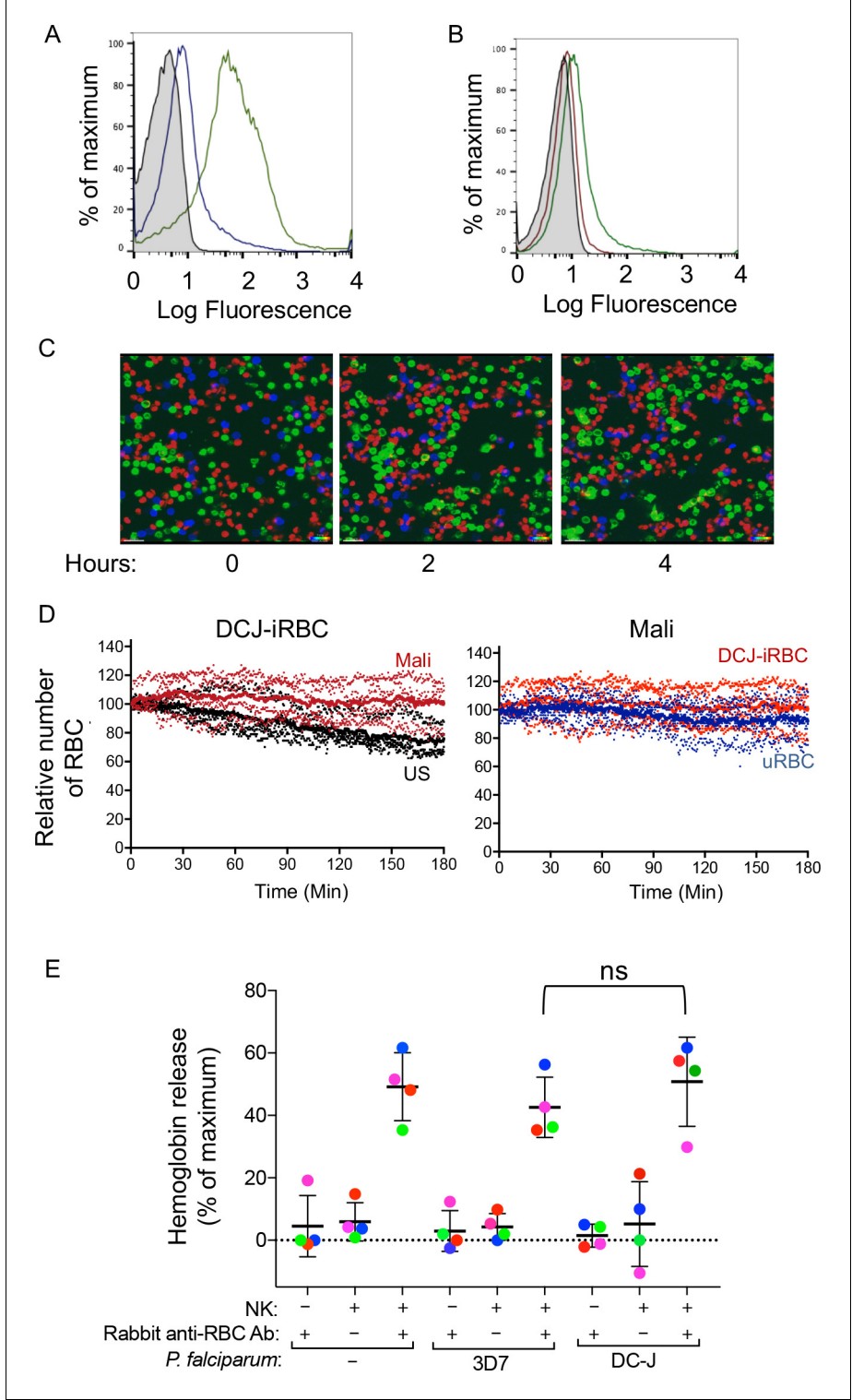

**Figure 3.** Naturally acquired antibodies to PfEMP1 have a predominant role in NK-mediated ADCC. (**A**) Immunostaining of uRBCs with Mali plasma (shaded) and of RBCs infected with *P.f.* 3D7 (green) or with *P.f.* DC-J parasites (blue). Bound Abs were detected with AF488-conjugated anti-human IgG (H + L) antiserum. (**B**) Staining of DC-J iRBCs with Abs from US serum (red) and Mali plasma (green). Secondary staining was as in (**A**). The shaded histogram represents staining with secondary Ab alone. (**C**) Live imaging of primary NK cells (green) co-incubated with uRBCs (blue) and DC-J-iRBCs (red) at an equal ratio (1:1:1) in the presence of US serum (1:10) or Mali plasma (1:10). Cell counts for NK cells, DC-J-iRBCs and uRBCs were determined every minute for 3 hr.

*Figure 3 continued on next page*

*Figure 3 continued*

Representative snapshots taken at time 0, 2 and 4 hr are shown. (D) Cell numbers were normalized to 100 at the start of image acquisition. Composite display of 4 independent experiments, each carried out with a different NK cell donor (dotted lines). The mean is shown as a solid line. (E) NK cell-mediated ADCC towards uRBCs, *P.f.* 3D7 iRBCs and *P.f.* DC-J iRBCs. Cells were mixed at an NK:RBC ratio of 5:1 and incubated for 5 hr in the presence or absence of rabbit anti-RBC serum (1:4000), as indicated. Hemoglobin release, measured using a Quantichrom Hb assay, is shown relative to release from RBCs treated with 1% Triton X-100. Data are shown (mean ± SD) for NK cells from four independent donors (t-test, p=0.3743, comparing 3D7 and DC-J in presence of NK cells and Rabbit anti RBC Ab).

DOI: https://doi.org/10.7554/eLife.36806.009

The following figure supplement is available for figure 3:

**Figure supplement 1.** Hemoglobin release assay.

DOI: https://doi.org/10.7554/eLife.36806.010

13.1% of total Hb content was released (*Figure 3E*). We concluded that uRBCs and 3D7 *P.f.*-iRBCs were equally sensitive to NK-mediated ADCC.

This approach gave us an opportunity to test whether DC-J-iRBCs were inherently resistant to NK-mediated ADCC. Lysis assays in the presence of rabbit anti-RBC Abs and NK cells were performed. The extent of Hb release (52.66 ± 11.34%) after incubation at an NK cell to DC-J-iRBC ratio of 5:1 for 5 hr was comparable to that obtained with uRBCs and 3D7-iRBCs (*Figure 3E*). Hemoglobin release in the absence of rabbit anti-human RBC Abs was minimal. Therefore, we concluded that the lack of lysis of DC-J-iRBCs in the presence of Mali plasma was not due to resistance to NK-dependent ADCC, but rather to the low amount of Abs bound to RBCs infected with this PfEMP1-deficient parasite strain. Together, these data suggested that naturally acquired Abs to PfEMP1 play a critical role in NK cell-mediated destruction of iRBCs.

## The FcγRIIIa binding site of human IgG1 Fc is required for NK-dependent lysis of *P.f.*-infected RBCs

Abs with broad reactivity against certain members of the RIFIN family of *P.f.* proteins have recently been cloned from memory B cells of malaria-exposed individuals in Kenya, Mali and Tanzania (*Pieper et al., 2017*; *Tan et al., 2016*). Similar to PfEMP1, RIFIN is a type of variant antigen expressed on the surface of iRBCs. Using a *P.f.* 3D7 strain enriched for expression of RIFIN (*Figure 4A*), we tested NK-dependent lysis of RIFIN⁺ iRBCs in the presence of the RIFIN-specific human monoclonal Ab MGD21. Lysis occurred during incubation with NK cells at an NK cell to iRBC ratio of 5:1 for 6 hr, as measured by Hb release (*Figure 4B*). Negligible lysis was observed in the absence of NK cells or in the absence of MGD21. We then tested a variant of monoclonal Ab MGD21 (MGD21-LALA), into which mutations had been introduced in the Fc to reduce binding to Fc receptors (*Tan et al., 2016*). Staining of iRBCs indicated that MGD21 and MGD21-LALA bound similarly to RIFIN⁺ iRBCs (*Figure 4A*). However, in the presence of NK cells, only MGD21, and not MGD21-LALA, induced Hb release (*Figure 4B*), demonstrating that an intact Fc receptor-binding site was required for NK cell stimulation. In addition, we concluded that *P.f.* antigens other than PfEMP1 have the potential to induce NK-dependent ADCC in the presence of specific Abs.The timing of eukaryotic evolution Does a relaxed molecular clock reconcile proteins and fossils?

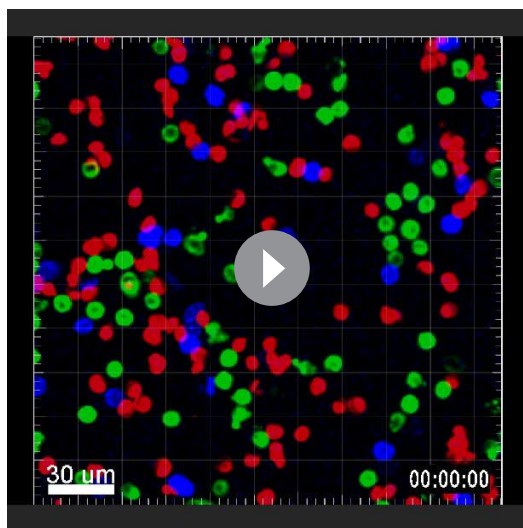

**Video 3.** shows a co-culture of NK cells, uRBC and RBC infected with the PfEMP1-deficient *P.f.* strain DC-J in the presence of Mali plasma, imaged for 106 min.

DOI: https://doi.org/10.7554/eLife.36806.011

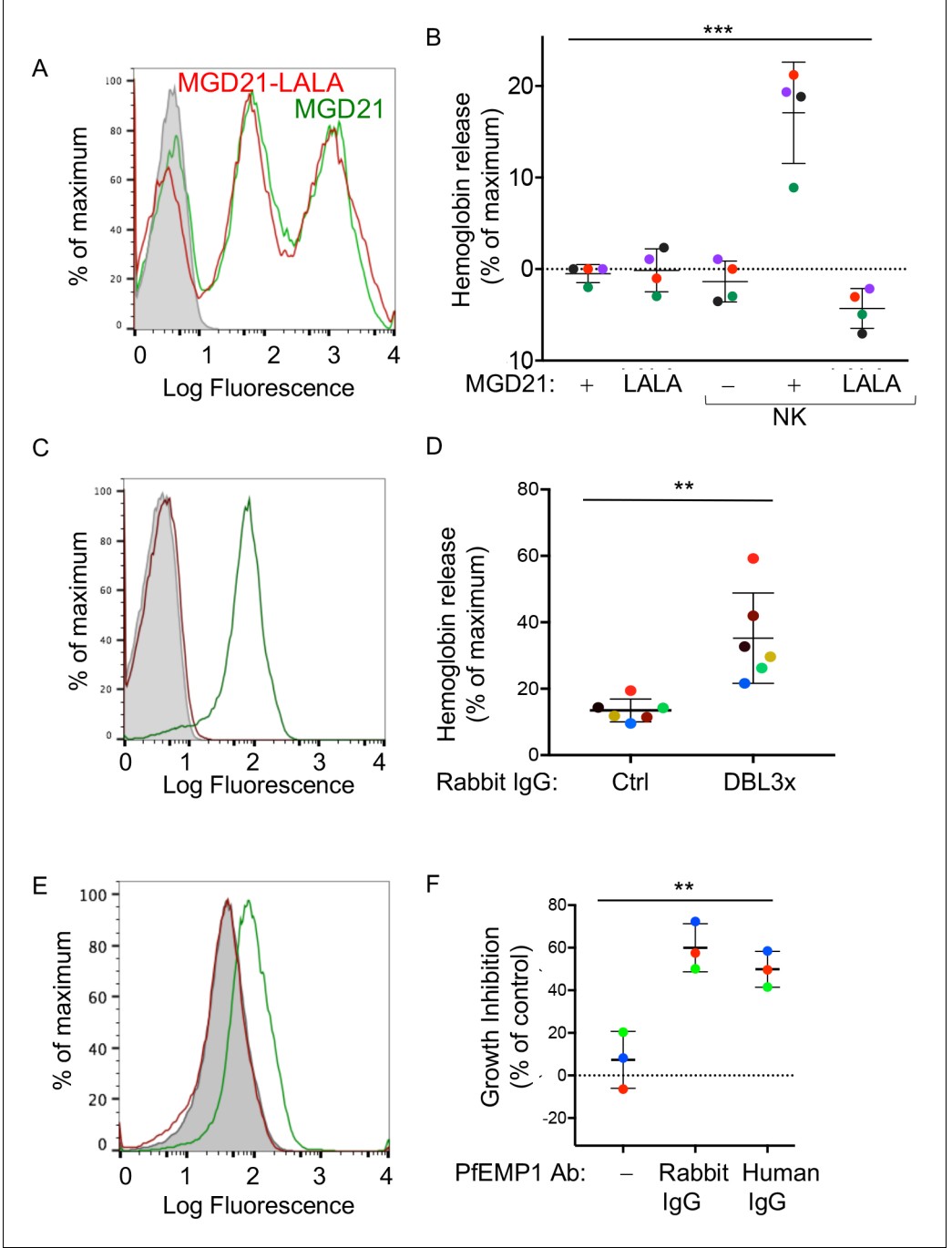

**Figure 4.** Human antibodies to RIFIN and to PfEMP1 promote NK-dependent lysis of iRBCs and inhibition of *P.f.* growth. (**A**) Staining of RBCs infected with a *P.f.* 3D7 strain enriched for expression of a RIFIN family member (PF3D7_1400600) with human monoclonal Ab MGD21 (green) or variant MGD21-LALA (red). The shaded histogram represents staining with AF488-conjugated anti-human IgG (H + L) antiserum alone. (**B**) Lysis of RIFIN[+] iRBCs incubated for 6 hr in the absence of NK cells or at an NK to iRBC ratio of 5:1 in presence or absence of MGD21 or MGD21 LALA Abs, as indicated. Data are shown (mean ± SD) for NK cells from four independent NK cell donors, as measured by Hb release (ANOVA, p=0.0005). (**C**) Staining of RBCs infected with *P.f.* FCR3 strain expressing VAR2CSA with rabbit polyclonal Ab to the DBL3X domain of PfEMP1 VAR2CSA (green) or with control rabbit serum (red). The shaded histogram represents staining with secondary FITC-labeled anti-rabbit IgG alone. (**D**) Hemoglobin release measured after incubation of NK cells with VAR2CSA-iRBCs, at an NK to iRBC ratio of 5:1 for 5 hr, in the presence of affinity-purified IgG from control rabbit serum or from serum of rabbit immunized with VAR2CSA PfEMP1. Each color represents a single NK cell donor tested in independent experiments (n = 6) (t-test,

*Figure 4 continued on next page*

*Figure 4 continued*
p=0.0049). (E) Staining of iRBCs expressing *P.f.* VAR2CSA with human polyclonal Abs to either AMA1 antigen as control (red), or to the DBL domains of PfEMP1 VAR2CSA (green). The shaded histogram represents staining with AF488-conjugated anti-human IgG (H + L) antiserum alone. (F) Parasite GIA-ADCC analyzed by flow cytometry. Enriched trophozoite-stage FCR3 VAR2CSA-iRBCs were incubated with NK cells, at an NK to iRBC ratio of 5 for 6 hr, in the presence of a control rabbit IgG (–), of rabbit anti-PfEMP1 IgG, and of human anti-PfEMP1 IgG, as indicated. A 100-fold excess of uRBCs (relative to the iRBCs input) was added, and incubation resumed for another 42 hr. Inhibition is expressed as percent decrease in parasitemia relative to iRBCs that were incubated with NK cells in the absence of Abs (ANOVA, p=0.0027).
DOI: https://doi.org/10.7554/eLife.36806.012
The following figure supplement is available for figure 4:

**Figure supplement 1.** PfEMP1 antibodies activate NK-dependent lysis of iRBCs and inhibition of *P.f.* growth.
DOI: https://doi.org/10.7554/eLife.36806.013

## Naturally acquired human IgG specific for pregnancy-associated VAR2CSA antigen promotes NK-dependent lysis of infected RBCs

To further define antigenic epitopes with the potential to induce NK-dependent ADCC toward *P.f.*-iRBCs, we tested polyclonal, affinity-purified IgG from rabbits that had been immunized with the Duffy binding-like 3x (DBL3X) domain of the PfEMP1 variant VAR2CSA (*Obiakor et al., 2013*). This rabbit IgG stained VAR2CSA-iRBCs, as measured by flow cytometry (*Figure 4C*). The VAR2CSA-specific rabbit IgG, but not control rabbit serum IgG, induced Hb release from VAR2CSA-iRBCs after incubation with NK cells (*Figure 4D*). Uninfected RBCs were not lysed in the presence of VAR2CSA-specific rabbit IgG (*Figure 4—figure supplement 1A*). These results showed that domain DBL3X was accessible to Abs at the surface of VAR2CSA-iRBCs, and oriented in such a way that bound Abs could engage FcγRIIIA on NK cells.

To test the potential of naturally acquired Abs to VAR2CSA PfEMP1 to promote NK-dependent inhibition of *P.f.* growth in RBCs, we used human IgG isolated from pooled plasma of multigravid women, and affinity-purified on DBL domains of VAR2CSA PfEMP1 (*Doritchamou et al., 2016*). This natural IgG stained VAR2CSA-iRBCs, as measured by flow cytometry, whereas human IgG Abs against another parasite antigen, AMA1, did not (*Figure 4E*). NK cells were incubated with trophozoite-stage enriched RBCs infected by *P.f.* VAR2CSA for 6 hr in the presence of IgG Abs. A 100-fold excess of uRBCs was added and incubation resumed for 42 hr. In the presence of naturally acquired human IgG specific for VAR2CSA PfEMP1, *P.f.* growth was inhibited by 49.88 ± 8.49%, which was similar to inhibition obtained with rabbit anti-DBL3X IgG (60 ± 11.29%) (*Figure 4F*, *Figure 4—figure supplement 1B*). No inhibition was observed in the absence of NK cells (*Figure 4—figure supplement 1C*). These results showed that naturally acquired Abs to PfEMP1 induce NK-mediated ADCC, which inhibits parasite growth in RBCs.

## Discussion

The main objective of our study was to test whether NK cells could help control blood-stage malaria by lysing iRBCs through ADCC. Considering the essential role of Abs in conferring clinical immunity to individuals living in areas of high *P.f.* transmission (*Cohen et al., 1961*), and the limited efficacy of malaria vaccines tested so far, any immune effector function that depends on Abs needs to be evaluated. We provide strong evidence of Ab-dependent NK cell cytotoxicity towards *P.f.*-iRBCs in the presence of Abs from malaria-exposed individuals in Mali. NK cell responses to iRBCs and their effect on *P.f.* growth in culture were tested using primary, unstimulated human NK cells. NK cells isolated from the blood of individuals in the US were used. It is possible that NK cells from individuals exposed to malaria may have altered function, or a reduced cell frequency, as reported following a controlled human malaria infection (*Mpina et al., 2017*). Lysis of iRBCs by NK cells, in the presence of Abs to *P.f.* antigens exposed at the surface of iRBCs, was highly selective, leaving most uRBCs intact. NK cell-mediated ADCC inhibited *P.f.* growth in RBC cultures. Human Abs specific for a single class of *P.f.* antigens expressed at the surface of RBCs, such as PfEMP1 and RIFIN, were sufficient to induce NK cell cytotoxicity and *P.f.* growth inhibition. We propose that NK-dependent ADCC may

be an effective mechanism to limit parasite growth, as it combines the powerful cytotoxicity of innate NK cells with the specificity of Abs generated by adaptive immunity.

RIFIN antigens may help *P.f.* evade immune detection by binding to inhibitory receptors on human lymphocytes (*Saito et al., 2017*). However, we observed hemoglobin release after mixing NK cells with RIFIN⁺ *P.f.*-infected RBCs in the presence of a human monoclonal Ab to RIFIN. NK cell activation was not due to the masking of RIFIN by the monoclonal Ab (and release from inhibition) but to activation of ADCC by the human IgG1 mAb because the same antibody lacking FcR binding did not activate NK cells.

The developmental cycle of *P.f.* in iRBCs provides a window of opportunity for Ab-dependent immune effector responses. Following merozoite invasion of RBCs, *P.f.* proteins begin to appear at the RBC surface after 16–20 hr and remain exposed until infectious merozoites are released. Once released, merozoites rapidly invade fresh RBCs (*Boyle et al., 2010*). Therefore, it is likely that high Ab titers are needed to neutralize merozoites and block entry into RBCs. In contrast, RBCs harboring non-infectious *P.f.*, as it progresses through trophozoite and schizont stages, display *P.f.* antigens at their surface for more than 24 hr, and ADCC responses activated by FcγRIIIa in primary human NK cells are rapid, strong and independent of coactivation signals (*Bryceson et al., 2005*).

Evidence of NK cell activation and RBC lysis was obtained with three different assays: (1) NK cell degranulation and cytokine production by flow cytometry, (2) loss of intact *P.f.*-iRBCs by live imaging, and (3) RBC lysis by measurement of Hb release. NK cell degranulation in a co-culture with *P.f.*-iRBCs, selectively induced by Abs from malaria-exposed individuals, was just as strong as that obtained with rabbit polyclonal antiserum raised against human RBCs. Furthermore, using Hb-release assays and rabbit anti-RBC serum, it was possible to show that iRBCs are not inherently more resistant or sensitive than uRBCs to NK-mediated lysis. Live imaging of a coculture of NK cells, uRBCs and *P.f.*-iRBCs, in the presence of plasma from malaria-exposed individuals, revealed selective lysis of iRBCs, with no 'bystander' lysis of uRBCs.

Natural cytotoxicity of NK cells towards *P.f.*-iRBCs was not detected in our assays with resting NK cells. A recent study in humanized mice reconstituted with human lymphocytes and injected with *P.f.*-infected human RBCs, reported some lysis of *P.f.*-iRBCs by NK cells (*Chen et al., 2014*). It is possible that under specific stimulatory conditions, including soluble factors and contact with other cells, human NK cells exhibit natural cytotoxic responses towards *P.f.*-iRBCs. However, considering that clinical immunity to malaria depends in large part on Abs, and that development of an effective vaccine is a high priority, we chose to focus on ADCC by NK cells. Signaling in NK cells by FcγRIIIa alone, independently of other co-activation signals and of integrin-dependent adhesion, is sufficient to induce strong responses, unlike other NK activation receptors, which require synergy through combinations of co-activation receptors (*Bryceson et al., 2005*; *2006b*). The Ab-mediated activation of NK cell cytotoxicity described here is adding a strong effector mechanism to the other mechanisms by which Abs may confer protection against malaria, including neutralizing Abs and Abs that activate the complement pathway (*Boyle et al., 2015*).

We have shown that NK cell-mediated ADCC inhibits the growth of *P.f.* in RBC cultures in the presence of Abs to *P.f.* antigens expressed at the surface of iRBCs in a standard growth inhibition assay (GIA) by co-incubation of iRBCs and NK cells with a large excess of uRBCs. To distinguish inhibition by NK cells from other Ab-dependent functions, such as merozoite neutralization and activation of complement, we developed a two-step GIA to evaluate inhibition that had occurred prior to iRBC rupture and reinvasion of fresh RBCs. As inhibition of *P.f.* growth occurred in the presence of purified IgG from plasma of malaria-exposed individuals, other serum components were not required for NK-mediated inhibition. The modified GIA for ADCC could help define *P.f.* antigens that induce Abs of sufficient titer and quality for FcR activation (e.g. IgG isotype, glycosylation). The GIA-ADCC is well-suited to large screens of plasma from subjects in malaria vaccine trials or in studies of naturally acquired immunity to malaria.

Previous work has shown that IL-2 produced by T cells following malaria infection or injection of a malaria vaccine activates IFN-γ production by NK cells (*Wolf et al., 2017*). In addition, *P.f.* infection activates IL-18 secretion by macrophages. Through IL-18 and direct contact with macrophages NK cells are activated to produce IFN-γ (*Baratin et al., 2005*; *Wolf et al., 2017*). In contrast, the effector functions of NK cells we describe here are independent of external signals, since unstimulated primary NK cells respond directly to activation by multivalent IgG Fc binding to FcγRIIIa (*Bryceson et al., 2005*). Experiments performed here used NK cells freshly isolated from human

blood, without stimulation prior to incubation with RBCs and Abs. In summary, we have shown that human NK cells have the potential to control *P.f.* parasitemia through IgG-dependent activation of NK cellular cytotoxicity, and thus contribute to protection from blood-stage malaria.

# Materials and methods

## Key resources table

| Reagent type or resource | Designation | Source | Identifier | Additional information |
|---|---|---|---|---|
| Antibody | Human IgG (H + L) Alexa Fluor 488 | Thermo Fisher Scientific | A-11013 | |
| Antibody | Goat anti Rabbit Ig, FITC | Southern Biotech, AL | 4010–02 | |
| Antibody | Anti human CD56 PE Cy5.5 | Beckman Coulter, CA | Clone N901, | |
| Antibody | Anti human CD235a FITC | Biolegend, CA | Clone H1264 | |
| Antibody | Anti human CD45 PE | Biolegend, San Diego, CA | Clone H130 | |
| Antibody | Anti human IFN-γ—APC | BD Biosciences | Clone B27 | |
| Antibody | Anti human TNF-α—PE | BD Bioscience | Clone 6401.1111, | |
| Antibody | Anti human CD107a-PE | BD Biosciences | Clone H4A3, | |
| Antibody | Rabbit polyclonal anti-human red blood cell antibody | Rockland Immunochemicals Inc. PA | 209–4139 | |
| Antibody | VAR2CSA human Ab | Patrick E Duffy | *Doritchamou et al., 2016* | |
| Antibody | AMA1 human Ab | Patrick E Duffy | *Doritchamou et al., 2016* | |
| Antibody | DBL3X rabbit Ab | David L Narum | *Obiakor et al. (2013)* | |
| Antibody | Control IgG rabbit | David L Narum | *Obiakor et al. (2013)* | |
| Chemical compound, drug | eFluor 670 dye | Thermo Fisher Scientific | eBioscience Cell Proliferation Dye eFluor 670 Catalogue no 65-0840-85 | |
| Chemical compound, drug | eFluor 450 dye | Thermo Fisher Scientific | eBioscience Cell Proliferation Dye eFluor 450 Catalogue no 65-0842-85 | |
| Chemical compound, drug | PKH 67 dye | SIGMA Aldrich | PKH67 Green Fluorescent Cell Linker Kit for General Cell Membrane Labeling Catalogue no. PKH67GL SIGMA | |
| Strain, strain background (*Plasmodium falciparum*) | 3D7 | Aaron T Neal | | |
| Strain, strain background (*Plasmodium falciparum*) | NF54 DC-J | Kirk W Deitsch | *Dzikowski et al. (2006)* | |
| Strain, strain background (*Plasmodium falciparum*) | FCR3 VAR2CSA | Patrick E Duffy | *Doritchamou et al. (2016)* | |
| Strain, strain background (*Plasmodium falciparum*) | 3D7-MGD21[+] | Antonio Lanzavecchia | *Tan et al. (2016)* | |
| Biological samples (Human) | Human peripheral blood mononuclear cells | Department of Transfusion Medicine, NIH | | |
| Biological samples (Human) | Serum | Valley Biomedical (Winchester, VA) | | |
| Biological samples (Human) | Plasma | Malaria Research and Training Centre, Bamako, Mali | *Crompton et al. (2008)* | |
| Software, algorithm | Imaris | Bitplane | http://www.bitplane.com | |
| Software, algorithm | Zen | Carl Zeiss | | |
| Software, algorithm | FlowJo | FlowJo, LLC | | |
| Software, algorithm | Graphpad PRISM 7.0 | GraphPad Software, Inc. | | |
| Commercial Assay or kit | Quantichrom hemoglobin assay kit | BioAssay Systems, CA | DIHB-250 | |

## Study approval

This study was approved by the Ethics Committee of the Faculty of Medicine, Pharmacy, and Dentistry at the University of Sciences, Techniques, and Technologies of Bamako, in Mali and by the Institutional Review Board of the National Institute of Allergy and Infectious Diseases, National Institutes of Health. Prior to inclusion in this study, written informed consent was received from participants.

## Antibody sources

Plasma samples were collected from adults enrolled in a multi-year malaria study in the rural village Kambila (*Crompton et al., 2008*), by starting with venous blood collected in citrate-containing cell-preparation tubes (BD, Franklin Lakes, NJ). Samples were transported 45 km away to the Malaria Research and Training Centre in Bamako, where peripheral blood mononuclear cells (PBMCs) and plasma were isolated. Plasma was frozen in 1 ml aliquots at −80°C. Samples were shipped to the United States on dry ice for analysis. Control serum from US individuals was obtained from Valley Biomedical (Winchester, VA).

## Enrichment of antibody

IgG was purified from plasma or serum by standard affinity chromatography. Briefly, each sample was diluted 1:5 in column equilibration-wash buffer (10 mM $NaPO_4$, 150 mM NaCl, pH 7.0). The IgG fraction was isolated on Protein G columns (GE Healthcare, Amersham-Pharmacia-HiTrap Protein G) and eluted with 100 mM glycine, pH 2.5. and immediately neutralized to pH 7.4 with 4.0 M Tris pH 8.0. IgG was concentrated and dialyzed in Pall Macrosep columns (30 kDa MW cutoff) with PBS.

## NK cell isolation

Human blood samples from deidentified healthy US donors were drawn for research purposes at the NIH Blood Bank under an NIH IRB approved protocol with informed consent. PBMCs were first isolated using Lymphocyte Separation Medium (MP Biomedicals, Solon, OH), washed with PBS twice, and resuspended in PBS, with 2% FBS and 1 mM EDTA. NK cells were isolated from PBMCs by depletion of non-NK cells using an NK cell isolation kit (STEMCELL Technologies, Cambridge, MA). The manufacturer's protocol was modified as follows. PBMCs were resuspended at $2 \times 10^8$, per ml and 2.5 μl/ml of anti-CD3 biotin (STEMCELL Technologies) was added to the 50 μl/ml of the non-NK cocktail to increase NK cell purity. Resting NK cells were resuspended in Iscove's modified Dulbecco's medium (IMDM; Invitrogen, Carlsbad, CA) supplemented with 10% human serum (Valley Biomedical, Winchester, VA), and were used within 1 to 4 days after isolation. Only NK cell preparations that had greater than 95% CD14neg, CD3neg, CD56pos, as determined by flow cytometry, were used in experiments.

## Cultivation and purification of *P. falciparum*

3D7 and FCR3 VAR2CSA strains were cultivated at 37°C under 5% $O_2$, 5% $CO_2$, 90% $N_2$ at 37°C at <5% hematocrit using $O^+$ human erythrocytes (Interstate Blood Bank, Inc.). The *P.f.* DC-J strain was cultivated similarly in the presence of Blasticidin (2.5 μg/ml). Parasites were cultured in complete medium, which was RPMI 1640 buffered with 25 mM HEPES and supplemented with 2.5% heat-inactivated human AB Serum, 0.5% Albumax-II, 28 mM sodium bicarbonate, 25 μg/ml gentamycin, and 50 μg/ml hypoxanthine. Parasite development was monitored by light microscopy using methanol-fixed, Giemsa-stained thin blood films. Parasites were synchronized using sorbitol (*Lambros and Vanderberg, 1979*). Parasite-iRBCs were enriched for knobs using Zeptogel (contains gelatin) sedimentation routinely. Infected RBCs used in ADCC assays were enriched at the trophozoite stage with percoll-sorbitol gradient and centrifugation (*Aley et al., 1984*; *Hill et al., 2007*), washed, and resuspended in complete medium in the absence of human serum. Cultures growing in Albumax-II therefore had no antibodies or complement components.

## Immunostaining and immunofluorescence analysis

iRBCs, enriched at the trophozoite stage, were resuspended in PBS and 2% FBS, and incubated at 4°C for 30 min with plasma, serum, or purified antibodies at specified dilutions. Cells were washed and incubated at 4°C with appropriate and fluorescently-tagged secondary Abs for an additional 20

min. Cells were washed, and flow cytometry was performed on a FACS LSR-II or a FACS Calibur (BD Biosciences), and data analyzed with FlowJo (FlowJo, LLC). For immunofluorescence analysis, iRBCs, enriched at the trophozoite stage, were mixed with uRBCs at a ratio of 1:1 and incubated with US serum or Mali plasma for 30 min at 4°C. Cells were washed and incubated with fluorescently –tagged secondary Abs for an additional 20 min. Cells were first washed, and then fixed with 1% paraformaldehyde (PFA) at room temperature. DAPI was used to visualize the *P.f.* DNA in iRBCs. Immunofluorescence images were obtained on a LSM 780 confocal laser microscope (Carl Zeiss, Oberkochen, Germany). Images were acquired using the Zen software (Carl Zeiss).

## Scanning electron microscopy

uRBCs and iRBCs were washed with 0.1 M phosphate buffer (pH 7.4). RBCs were centrifuged at 2500 rpm for 5 min and supernatants were removed. RBCs were again washed with 0.1 M phosphate buffer (pH 7.4). RBCs were resuspended in 3 ml of fixative solution (3% PFA +0.1% glutaraldehyde). The cells were stored at 4°C until further processing for imaging. Fixed cells were allowed to settle on silicon chips for 1 hr. Subsequent post-fixation with 1% $OsO_4$ was performed with microwave irradiation (Pelco 3451 microwave processor, in cycles of 2 min on, 2 min off, 2 min on at 250 W under 15 in. Hg vacuum; Ted Pella, Redding, CA). Specimens were dehydrated in a graded ethanol series for 1 min under vacuum. Samples were then dried to a critical point in a Bal-Tec cpd 030 drier (Balzer, Bal-Tec AG, Balzers, Liechtenstein). Cells were then coated with 75 Å of iridium in an IBS ion beam sputter (South Bay Technology, Inc., San Clemente, CA.) Samples were imaged on a Hitachi SU-8000 SEM (Hitachi, Pleasantown, CA).

## Degranulation and intracellular cytokine assays

Resting NK cells ($2 \times 10^5$) either alone or mixed with enriched iRBCs ($2 \times 10^5$) were added to wells of a 96-well plate, in the presence of antibodies (either US serum diluted 1:10, Mali plasma diluted 1:10, or rabbit anti-human RBC antibody at 1.25 µg/ml). Anti-CD107a Ab–PE diluted 1:20 (Clone H4A3, Cat#555801, BD Biosciences, Franklin Lakes, NJ) was added at the beginning of the incubation with NK cells. Cells were centrifuged for 3 min at 100 g and incubated for 4 hr at 37°C. Cells were centrifuged, resuspended in PBS containing 2% FBS, and stained with conjugated anti-CD56–PC5.5 Ab (Clone N901, Beckman Coulter, Brea, CA), near-IR fixable Live/Dead dye (Invitrogen), and CD235a–FITC (Clone H1264, Biolegend, San Diego, CA). Samples were analyzed on a FACS-LSRII flow cytometer (BD Biosciences) and data analyzed with FlowJo software (FlowJo, LLC). For intracellular cytokine assays, NK cells were incubated with iRBCs as described above in the presence of Brefeldin A (5 µg/ml) during the 4 hr incubation. Cells were then stained with anti-CD56–PC5.5 Ab (Clone N901, Beckman Coulter, Brea, CA), near-IR fixable Live/Dead dye (Invitrogen), CD235a–FITC (Clone H1264, Biolegend, San Diego, CA), followed by fixation and permeabilization using the BD Cytofix/Cytoperm Kit (BD Biosciences). IFN-γ was detected using anti-IFN-γ–APC Ab (Clone B27, BD Biosciences) and TNF-α was detected using anti-TNF-α–PE Ab (Clone 6401.1111, BD Bioscience). Samples were resuspended in PBS and analyzed on a FACS LSRII flow cytometer (BD Biosciences). Data analysis was performed with FlowJo software (FlowJo, LLC).

## Time-lapse imaging

NK cells, uRBCs and iRBCs were washed twice with PBS before labeling with different dyes. iRBCs were stained with cell proliferation dye eFluor 670 at 5 µM for 5 min in PBS at 37°C. Similarly, uRBCs were stained with eFluor 450 dye at 2.5 µM concentration for 5 min in PBS at 37°C. NK cells were washed, suspended in diluent C and stained with 1 µM PKH67 membrane dye (PKH67 green fluorescent green linker kit, Sigma-Aldrich) for 5 min at 37°C. Cells were then washed three times with media containing serum (e.g., RPMI with 10% FBS). For imaging, cells were resuspended in RPMI 1640 containing 0.5% Albumax-II in the absence of Phenol Red. Cells were added in 8-well Lab-Tek I Chambered cover glass (Nunc) and allowed to settle for 15 min. Imaging was performed with a Zeiss LSM 780 confocal microscope while maintaining incubation condition at 37°C, 5% $CO_2$, in a humidified chamber. Images were acquired at 30 s interval for 6 hr. Time-lapse image stacks were imported into the Imaris software. A threshold algorithm eliminated background noise from each channel, and a Gaussian filter was applied to smooth the texture, and to easily segment the regions of interest (ROIs). After filtering, a surface channel was created from each color channel for each cell

population, with surface threshold based on intensity. The surface generator was set to run a watershed algorithm. The seed-points diameter was set to 4.5 μm for iRBCs and uRBCs, and 6.0 μm for NK cells. In order to weed out unwanted particles that passed the intensity threshold, a surface ROI was considered to be one with voxel size greater than 110 voxels. For the tracking algorithm we used autoregressive motion with maximum step distance set to 5 μm and maximum gap size set to two frames.

## Growth inhibition assay (GIA)

NK cells were incubated with $20 \times 10^3$ trophozoite-stage iRBCs at NK to iRBC ratios of 1:1 and 3:1 in the presence of $20 \times 10^5$ uRBCs in 96-well plates for 48 hr at 37°C, in complete medium. Thin blood smears were fixed in 100% methanol, stained with 5% Giemsa solution and counted under light microscope. 25 microscope fields, each containing 200 RBCs, were counted. Parasitemia was expressed as [(number of iRBCs ÷ total number of RBCs)×100]. $2.5 \times 10^5$ NK cells and $5 \times 10^4$ FCR3 VAR2CSA–iRBCs were mixed in 96-well plates and incubated for 6 hr at 37°C in the absence or presence of purified rabbit IgG antibodies to the DBL3X domain (0.5 mg/ml), or purified human IgG antibodies to the DBL domains of PfEMP1 VAR2CSA (0.5 mg/ml), or control rabbit IgG (0.5 mg/ml) in a final volume of 25 μl. A 100-fold excess of uRBCs ($5 \times 10^6$) was then added, bringing the final volume to 100 μl. Cultures were then maintained for an additional 42 hr at 37°C in standard parasite growth conditions. At the end of incubation, CD45-PE (Clone H130, Biolegend, San Diego, CA) and CD235a-FITC antibodies, and Hoechst were used to stain NK cells, uRBCs and iRBCs. Samples were acquired on FACSLSR-II, and data analyzed with FlowJo (FlowJo, LLC). Parasitemia was determined as the fraction of RBCs (CD235a$^+$) positive for Hoechst. Samples with NK cells but in the absence of antibodies were used as control to calculate growth inhibition.

## GIA-ADCC assay

NK cells and iRBCs were resuspended in experimental media (no human serum). $6 \times 10^5$ NK cells and $2 \times 10^5$ iRBCs were mixed at a 3:1 ratio in 96-well plates and incubated for 6 hr at 37°C in the absence or presence of antibodies. For experiments using plasma or serum, the total amount of plasma or serum in each condition was (20 μl plasma/serum into 200 μl media) to control for the level of plasma. 20 μl of US serum the negative control, then increasing volume of Mali immune plasma was added in (Example: 2 μl Mali plasma with 18 μl US serum totaling 20 μl plasma/serum). After a 6 hr coincubation of iRBCs and NK cells, soluble Abs were removed by a wash. This washing step removed any antibody that would bind to merozoites. A 100-fold excess of uRBCs ($2 \times 10^7$) relative to iRBCs was then added and cultures were maintained for an additional 16 hr at 37°C in standard parasite growth conditions. At the end of incubation, CD45- PE (BD Biosciences), CD235a-FITC, and Hoechst were used to stain NK cells, uRBCs and iRBCs (*Figure 2—figure supplement 1C*). Samples with NK cells but in the absence of antibodies were used as control to calculate growth inhibition.

## Hemoglobin release assay

Enriched iRBCs and NK cells were washed with RPMI 1640, containing 0.5% Albumax in the absence of Phenol red. Cells were mixed at defined ratios in 96-well V bottom plates in 150 μl. Antibodies were added as specified. Antibodies tested in the assay are Rabbit anti-human RBC antibodies (1.25 μg/ml), MGD21 and MGD21-LALA antibodies (*Tan et al., 2016*) (0.2 mg/ml), Rabbit VAR2CSA and Control IgG antibodies (0.5 mg/ml). Cells were centrifuged at 100 g for 3 min and incubated at 37°C for 5–6 hr as mentioned. Plates were centrifuged at 2000 rpm for 5 min and 50 μl of supernatant was collected. Hemoglobin was measured with QuantiChrom Hemoglobin Assay Kit (BioAssay, Hayward, CA). Hemoglobin absorbance was measured at 405 nm using a 96-well plate reader (Enspire, Perkin Elmer, MA and SpectraMax plus, Molecular Devices, CA). In each experiment, maximum hemoglobin release was determined by lysis of iRBCs in 1% Triton-X-100. At the end of the 5 hr incubation period, the hemoglobin released in supernatant was quantified as percent fraction of maximum hemoglobin release. Hemoglobin released during the 5 hr incubation period in iRBCs sample was subtracted in each experiment to normalize the background in all experiments.

## Statistical analysis

Each graph was generated from at least three independent experiments. For normally distributed data, either mean ± SD or mean ± SEM was used, as specified. Statistical analysis was performed using the software Graphpad Prism v7. Data was analyzed by either two-tailed Student's t test, or by one-way analysis of variance (ANOVA).

## Acknowledgements

We thank AT Neal for *P.f.* 3D7 strain enriched for knobs, SA Arredondo for help with *P.f.* parasite culture, KW Deitsch for *P.f.* DC-J strain, A Lanzavecchia for Abs MGD21 and MGD21-LALA, and *P.f.* 3D7 strain enriched for RIFIN expression, B Hansen for scanning electron microscopy, L Lantz for IgG purification, J Skinner for statistics, and A Sajid for comments on the manuscript.

## Additional information

### Competing interests

L Michael Thomas: L.M. Thomas is an employee of Pfizer Inc, with ownership of stocks in Pfizer Inc. The other authors declare that no competing interests exist.

### Funding

| Funder | Grant reference number | Author |
| --- | --- | --- |
| National Institute of Allergy and Infectious Diseases | Z01 AI000525-30 LIG | Eric O Long |

The funders had no role in study design, data collection and interpretation, or the decision to submit the work for publication.

### Author contributions

Gunjan Arora, Geoffrey T Hart, Conceptualization, Methodology, Investigation, Formal analysis, Writing—original draft; Javier Manzella-Lapeira, Methodology, Investigation; Justin YA Doritchamou, David L Narum, L Michael Thomas, Ogobara K Doumbo, Boubacar Traore, Resources; Joseph Brzostowski, Methodology; Sumati Rajagopalan, Methodology, Writing—review and editing; Louis H Miller, Conceptualization, Methodology; Susan K Pierce, Conceptualization, Funding acquisition; Patrick E Duffy, Resources, Writing—review and editing, Funding acquisition; Peter D Crompton, Conceptualization, Resources, Writing—review and editing, Funding acquisition; Sanjay A Desai, Methodology, Writing—review and editing, Supervision, Funding acquisition; Eric O Long, Conceptualization, Methodology, Writing—original draft, Writing—review and editing, Supervision, Funding acquisition

### Author ORCIDs

Gunjan Arora (ID) http://orcid.org/0000-0003-0575-6205
Susan K Pierce (ID) http://orcid.org/0000-0001-7261-3437
Patrick E Duffy (ID) https://orcid.org/0000-0002-4483-5005
Sanjay A Desai (ID) http://orcid.org/0000-0003-2150-2483
Eric O Long (ID) http://orcid.org/0000-0002-7793-3728

### Ethics

Human subjects: The Ethics Committee of the Faculty of Medicine, Pharmacy and Dentistry at the University of Sciences, Technique and Technology of Bamako, and the Institutional Review Board of the National Institute of Allergy and Infectious Diseases, National Institutes of Health, approved this study. Written informed consent was obtained from all participants prior to inclusion in this study. Peripheral blood samples from healthy Malian adults enrolled in NIAID-NIH protocol # 07-I-N141 were analyzed. Peripheral blood samples from healthy U.S. adults were obtained from the NIH

Department of Transfusion Medicine under an NIH Institutional Review Board-approved protocol (99-CC-0168) with informed consent.

## Decision letter and Author response
Decision letter https://doi.org/10.7554/eLife.36806.016
Author response https://doi.org/10.7554/eLife.36806.017

## Additional files
### Supplementary files
• Transparent reporting form
DOI: https://doi.org/10.7554/eLife.36806.014

### Data availability
All data generated or analysed during this study are included in the manuscript and supporting files.

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
