## [Decision Letter]

Thank you for submitting your article "NK cells inhibit *Plasmodium falciparum* growth in red blood cells via antibody dependent cellular cytotoxicity" for consideration by *eLife*. The manuscript was reviewed by two reviewers and a member of the Board of Reviewing Editors with expertise in NK cells and malaria. The evaluation has been overseen by Tadatsugu Taniguchi as the Senior Editor. The following individual involved in review of your submission has agreed to reveal her identity: Megan Cooper (Reviewer #2).

In general, the paper was well received but there are some minor issues that should be addressed in a revised manuscript. The full reviews are given below. Please address the concerns in a rebuttal document and in your revised manuscript. We look forward to receiving your manuscript soon.

This manuscript provides a strong case for human NK cells being activated through ADCC against antibody-coated malaria infected RBCs. There is no evidence for spontaneous cytotoxicity of NK cells against infected RBCs in the absence of antibodies, and the effect requires use of serum from patients in a malaria-endemic area. The studies are well done and convincing. Although it is difficult to know how important this mechanism is in actual control of human malaria infection in vivo, such an assessment would require in vivo human studies that are not feasible. On the other hand, this appears to be an important conceptual advance to guide the development of effective malaria vaccines.

*Reviewer #1:*

This manuscript provides a strong case for human NK cells being activated through ADCC against antibody-coated malaria infected RBCs. There is no evidence for spontaneous cytotoxicity of NK cells against infected RBCs in absence of antibodies, and the effect requires use of serum from patients in a malaria-endemic area. The studies are well done and convincing. The primary issue is that it is difficult to know how important this mechanism is in actual control of malaria infection in vivo but that would require in vivo human studies that are not feasible. On the other hand, this appears to be an important conceptual advance to guide the development of effective malaria vaccines.

It would be of interest to know if the KIR phenotype of the human NK cells is related to response to infected RBCs, and to the corresponding HLA alleles of the NK cells. But this is beyond the scope of the current paper.

*Reviewer #2:*

Long and colleagues investigate whether NK cells can recognize and target *Plasmodium falciparum* infected RBCs by ADCC. They use a complementary assay, including a newly developed GIA-ADCC assay, to interrogate the ability of NK cells to kill infected cells. Their data clearly demonstrates that NK cells can mediate ADCC by naturally occurring serum and IgG from infected patients. They demonstrate that this can be mediated by naturally-occurring antibodies to PfEMP1 and VAR2CSA, as well as with a monoclonal antibody recognizing the RIFIN antigen. The authors conclude that NK-mediated recognition of infected RBCs and inhibition of parasite growth is an important anti-malarial mechanism to investigate when considering vaccine strategies.

Overall, experiments are well-controlled, and the hypotheses and results are clear and well-written. There has been some conflicting information in the field with regards to the potential for NK cells to recognize infected RBCs, and these studies using freshly isolated NK cells and serum from infected patients should clarify this.

There was a recent publication describing RIFIN proteins binding to inhibitor receptors as a mechanism of immunoevasion (Saito et al., 2017). The authors use a monoclonal ab against RFIN and a control with a mutation in the Fc domain to demonstrate specificity for ADCC. While the controlled study here demonstrates that the NK cytotoxic effect is most likely due to impaired activation, rather than blocking an inhibitory receptor ligand, a discussion and reference of this recent manuscript seems appropriate.

*Reviewer #3:*

In this manuscript Arora et al. report that human NK cells can efficiently lyse red blood cells infected by *Plasmodium falciparum* via antibody-dependent cellular toxicity.

Overall, the authors present a set of well-designed experiments that are easy to interpret, the data support their conclusions and form a cohesive story on a subject matter that is of interest to a broad audience.

I have a few comments that I think will help to further strengthen the manuscript:

- The authors use US serum, but Mali plasma: I think it would be useful to explain in the discussion why serum was used for the US control group and whether this difference could at all affect the experimental outcome (increase or decrease ADCC)

- The authors use NK cells from naïve US donors and conclude that NK "thus contribute to protection from blood-stage malaria": a recent report by Mpina et al., (2017) shows that NK cell frequency and numbers decrease in the blood following (and possibly during) blood stage parasitemia indicating that NK cells present in the blood during the blood stage of infection may differ in number, composition and function from NK cells present in the blood of a healthy cohort. The notion that the NK compartment could be different in infected vs. naïve cohorts (potentially affecting ADCC) should be acknowledged in the discussion.

- I do not have the expertise to fully evaluate the parasite strains used in Figure 3 and Figure 4, but I am wondering if overall parasite protein expression in iRBCs could change with the deletion or enhanced expression of the protein of interest (which would affect the interpretation of the data)

- Statistical analysis: please indicate in each figure legend when a t test vs. anova was done.

---

## [Author Response]

Reviewer #1:

*This manuscript provides a strong case for human NK cells being activated through ADCC against antibody-coated malaria infected RBCs. There is no evidence for spontaneous cytotoxicity of NK cells against infected RBCs in absence of antibodies, and the effect requires use of serum from patients in a malaria-endemic area. The studies are well done and convincing. The primary issue is that it is difficult to know how important this mechanism is in actual control of malaria infection* in vivo *but that would require* in vivo *human studies that are not feasible. On the other hand, this appears to be an important conceptual advance to guide the development of effective malaria vaccines.*

It would be of interest to know if the KIR phenotype of the human NK cells is related to response to infected RBCs, and to the corresponding HLA alleles of the NK cells. But this is beyond the scope of the current paper.

It would be interesting to examine the impact of HLA and KIR genes on susceptibility to malaria. One question is whether NK cell licensing through inhibitory KIR makes for stronger ADCC responses to infected RBC. It could be tested by multi-color flow analysis of NK cell phenotypes and ADCC function, preferably with samples from malaria-exposed individuals. One difficulty here is the extensive polymorphism of KIR genes, which is particularly high in Africa. Standard antibodies to specific members of the KIR family coul miss some of the polymorphic alleles. Nevertheless, such a study is feasible. A phenotypic and functional analysis of NK cells with samples from Africa would have to be done with priority given to KIR and NKG2A phenotypes, as this alone would require a fairly large number of antibodies. We have not performed such a study yet. As mentioned by the reviewer, it would be outside the scope of this paper.

Another question is whether specific combinations of KIR and HLA haplotypes are associated with resistance to malaria. It should be studied with large cohorts of individuals for which good clinical data and genetic analysis of their HLA and KIR genes are available. This specific question, which is not related to the main topic of our manuscript, has not been addressed on a large scale yet.

Reviewer #2:

Long and colleagues investigate whether NK cells can recognize and target Plasmodium falciparum infected RBCs by ADCC. They use a complementary assay, including a newly developed GIA-ADCC assay, to interrogate the ability of NK cells to kill infected cells. Their data clearly demonstrates that NK cells can mediate ADCC by naturally occurring serum and IgG from infected patients. They demonstrate that this can be mediated by naturally-occurring antibodies to PfEMP1 and VAR2CSA, as well as with a monoclonal antibody recognizing the RIFIN antigen. The authors conclude that NK-mediated recognition of infected RBCs and inhibition of parasite growth is an important anti-malarial mechanism to investigate when considering vaccine strategies.Overall, experiments are well-controlled, and the hypotheses and results are clear and well-written. There has been some conflicting information in the field with regards to the potential for NK cells to recognize infected RBCs, and these studies using freshly isolated NK cells and serum from infected patients should clarify this.There was a recent publication describing RIFIN proteins binding to inhibitor receptors as a mechanism of immunoevasion (Saito et al., 2017). The authors use a monoclonal ab against RFIN and a control with a mutation in the Fc domain to demonstrate specificity for ADCC. While the controlled study here demonstrates that the NK cytotoxic effect is most likely due to impaired activation, rather than blocking an inhibitory receptor ligand, a discussion and reference of this recent manuscript seems appropriate.

We have now mentioned this interesting paper and the potential of NK cell inhibition by *Plasmodium* RIFIN proteins. However, RIFIN does not bind the wild-type LAIR-1 that is expressed on NK cells. Furthermore, we did observe hemoglobin release after mixing NK cells with *P.f*.-infected RBCs that had been selected for high expression of RIFIN, in the presence of a human monoclonal Ab to RIFIN. We can be certain that NK activation was not due to the masking of RIFIN by the mAb (and release from inhibition) but to activation of ADCC by the human IgG1 mAb because the same antibody lacking FcR binding did not activate NK cells. This is now discussed in the Discussion section of the revised manuscript.

Reviewer #3:

In this manuscript Arora et al., report that human NK cells can efficiently lyse red blood cells infected by Plasmodium falciparum via antibody-dependent cellular toxicity.Overall, the authors present a set of well-designed experiments that are easy to interpret, the data support their conclusions and form a cohesive story on a subject matter that is of interest to a broad audience.I have a few comments that I think will help to further strengthen the manuscript:- The authors use US serum, but Mali plasma: I think it would be useful to explain in the discussion why serum was used for the US control group and whether this difference could at all affect the experimental outcome (increase or decrease ADCC)

In the Mali cohort, blood samples are anticoagulated to allow for the isolation of both peripheral blood mononuclear cells (PBMCs) and plasma. Serum pooled from multiple U.S. donors was obtained from Valley Biomedical and used as control. The main advantage of using a pooled serum lot is that it provides continuity in our experiments and does not introduce the variability inherent to individual human donors. To directly address the point raised by the reviewer, we purified IgG Abs from Mali plasma and U.S. serum. These purified IgG gave results similar to those obtained with plasma and serum, respectively. Specifically, IgG from U.S. controls did not bind to infected RBCs and did not activate NK cells, whereas IgG from Malian individuals bound to infected RBCs and inhibited parasite growth in a growth inhibition assay for ADCC (GIA-ADCC). A sentence has been added to the Results section.

- The authors use NK cells from naïve US donors and conclude that NK "thus contribute to protection from blood-stage malaria": a recent report by Mpina et al., (2017) shows that NK cell frequency and numbers decrease in the blood following (and possibly during) blood stage parasitemia indicating that NK cells present in the blood during the blood stage of infection may differ in number, composition and function from NK cells present in the blood of a healthy cohort. The notion that the NK compartment could be different in infected vs. naïve cohorts (potentially affecting ADCC) should be acknowledged in the discussion.

Our complete sentence was "we have shown that human NK cells have the potential to control P.f. parasitemia through IgG-dependent activation of NK cellular cytotoxicity, and thus contribute to protection from blood-stage malaria". The point raised by the reviewer has been discussed and the Mpina et al. paper has been cited in the Discussion section of the revised manuscript. In a separate, unpublished study, we have examined NK cells in a cohort of individuals living in a malaria-endemic region. In general, NK cells had good functionality, including ADCC toward infected RBCs. Given the small amount of blood collected from these individuals, it would not have been possible to use it to perform the experiments described in the present manuscript. Although we are not mentioning these unpublished results in the revised manuscript, they give us confidence that our results are relevant to disease.

- I do not have the expertise to fully evaluate the parasite strains used in Figure 3 and Figure 4, but I am wondering if overall parasite protein expression in iRBCs could change with the deletion or enhanced expression of the protein of interest (which would affect the interpretation of the data)

Figure 3 shows data obtained with the *P.f*. strain DC-J, which lacks expression of PfEMP1.

As a result, most of the reactivity with Abs from Mali adults was lost with DC-J-infected RBCs. PfEMP1 is known to be a dominant parasite protein expressed at the surface of infected RBC. The point of these experiments was to purposely change the overall amount of parasite PfEMP1 on iRBC. Experiments in Figure 4 used iRBC enriched for expression of RIFIN, or RBC infected with a strain that expresses the var2CSA variant of PfEMP1. Since these experiments were carried out using Abs specific to each of these antigens, either RIFIN or var2CSA, they were not intended to sample overall parasite proteins on iRBC.

- Statistical analysis: please indicate in each figure legend when a t test vs. anova was done

This has been done.